# A Strange Case of Traumatic Pleural Effusion: Pleural Empyema Due to *Actinomyces meyeri*, a Case Report

**DOI:** 10.3390/life13071450

**Published:** 2023-06-27

**Authors:** Marco Ghisalberti, Chiara Madioni, Giacomo Ghinassi, Uberto Maccari, Roberto Corzani, Fabiola Meniconi, Raffaele Scala, Piero Paladini

**Affiliations:** 1Thoracic Surgery Unit, Department of Medical, Surgical and Neuroscience Sciences, University Hospital of Siena, 53100 Siena, Italy; 2Pulmonology and Respiratory Intensive Care Unit, San Donato Hospital, 52100 Arezzo, Italy

**Keywords:** pulmonary actinomycosis, rare lung infection, *Actinomyces meyeri*, case report

## Abstract

BACKGROUND: Actinomycosis by *Actinomyces meyeri* is rare and scarcely reported in the literature. The lung is the main organ involved. Penicillin and amoxicillin are the first-choice treatments. Surgery is indicated when empyema and abscesses are resistant to medical treatment. CASE PRESENTATION: We report an underdiagnosed case of pleural empyema due to *A. meyeri* in a patient with closed chest trauma. The patient, a male, 47 years old, presented with a dry cough, thoracic pain, and dyspnea a month after the trauma. A chest X-ray showed a left lower lobe pleural effusion, so he was subjected to a thoracentesis, leading to a partial re-expansion of the left lung. The patient also complained about gum discomfort; thus, a dental x-ray scan was taken, which showed the presence of vertical bone resorption in a periodontal pocket. The patient was treated with levofloxacin 500 mg orally once a day, which was continued for 15 days after discharge. Two months after the accident, he presented again with intermittent fever, a worsening cough, and dyspnea. A CT scan showed thickening of the left pleura and a loculated pleural effusion with partial collapse of the left lower lobe. A decision was made to refer the patient to the Thoracic Unit to undergo surgery via a left thoracoscopic uniportal approach. The lung was thoroughly decorticated, and the purulent fluid was aspirated. The postoperative course was uneventful. Cultures showed the growth of *Actinomyces meyeri,* which is sensitive to imipenem and amoxicillin. The patient started a proper antibiotic regimen and, whenever possible, was discharged. At 12 months follow-up, a chest X-ray showed a complete resolution of the left pleural effusion with complete re-expansion of the left lung. CONCLUSIONS: Although rare, Actinomycetes infections must be considered especially in front of non-solving empyema or severe pneumonia of unknown cause because in the majority of cases, with the proper treatment, the restitutio ad integrum is possible.

## 1. Introduction

Actinomycosis is a rare but serious bacterial infection that affects various parts of the body, including the mouth, throat, lungs, and abdomen. *Actinomyces meyeri* is a specific species of Actinomyces bacteria that can cause actinomycosis in humans, and it is scarcely reported in the literature [1,2]. The first human case of infection by a bacterium of the genus *Actinomyces* was reported in the late nineteenth century. Since then, only a handful of cases have been described. Therefore, the disease spectrum remains mostly unclear [1,2]. Hence, we report a case of an initially unrecognized pleural empyema due to *A. meyeri* that developed as a late complication of a blunt chest trauma. We will explore the causes, symptoms, diagnosis, and treatment of actinomycosis sustained by *A. meyeri*, briefly discuss the risk factors for this condition, and provide some tips for preventing its onset.

## 2. Patient Information

We report the case of a 47-year-old male Caucasian patient. He had no significant past medical history, especially regarding gingival infections or dental diseases. He was a mild smoker (about 1 pack per day, 24 pack-years) with no history of alcohol or drug abuse. In August 2013, he was involved in a car accident. The patient suffered blunt chest trauma and a blow to the left arm. He was brought to the local hospital, where an X-ray showed the presence of a compound fracture of the left radius; a chest X-ray showed minor rib fractures without signs of hemothorax. No other site of trauma or pathological alteration was highlighted on that occasion. The patient recurred to the emergency department of a peripheral hospital nearly a month after the trauma. He complained about a persistent dry cough, pain in the left low thoracic region, and mild exertional dyspnea and shortness of breath.

On examination, the patient had reduced breath sounds on the left side of the chest. The patient was slightly hypoxic and hypocapnic, with arterial blood gas analysis in room air showing PaO_2_ = 66 mmHg, PaCO_2_ = 33.8 mmHg, and pH = 7.47. His other laboratory results were within normal limits, except for mild leucocytosis (WBC = 11.400/mm^3^) with neutrophils of 8.410/mm^3^. His kidney and liver function tests were normal. A chest X-ray revealed a left-sided pleural effusion with lower lobe atelectasis. He was then subjected to a diagnostic and therapeutic thoracentesis, which revealed the classic chronic hemorrhage. More than 800 mL of no longer clotting, old, and bloody pleural effusion was drained through a small-bore 18 French catheter. A chest X-ray was conducted shortly after the procedure. The X-ray showed only a partial re-expansion of the left lung. In order to address the residual left lower lobe atelectasis, bronchoscopy was performed. The endoscopic exam showed a modest quantity of mucopurulent, yellowish-white secretions on a background of substantially undamaged bronchial mucosa and the narrowing of B10 due to external compression. Microbiological examination of bronchoalveolar lavage was negative for pathogens. During the hospitalization, the patient also started complaining about gum discomfort with bleeding and pain, so a dental X-ray scan was taken. The exam showed the presence of vertical bone resorption in a periodontal pocket. According to the diagnostic hypothesis of a pleuro-pulmonary complication of the recent chest trauma, the patient was treated with levofloxacin 500 mg orally once a day during the peripheral hospital stay; this medication regimen was continued for 15 days after discharge. Two months after the accident, after a quick assessment at our emergency department, the patient was referred to our pulmonology department. He described a clearly worsened clinical picture with intermittent fever, a worsened dry cough, exertional dyspnea, and two episodes of hemoptysis. This time, his labs showed mild anemia (HB 12.4 g/dL) and an elevated level of CRP (15.3 mg/dL). Blood, stool, and urine samples were collected in order to address the source of infection, but all the cultures resulted negative. The chest X-ray showed thickening of the left pleura and an increased pleural effusion (Figure 1a). A CT scan showed thickening of the left pleura (split pleura sign) and a loculated pleural effusion with partial collapse of the left lower lobe (Figure 1b). Given the patient’s young age, delayed presentation, and worsening imaging findings, including the presence of loculations, a decision was made to refer the patient to the Thoracic Surgery Unit of Siena University Hospital to be considered for surgery. After a collegial discussion of the clinical case, the radiological evidence, and the anamnesis, a left thoracoscopic approach with a single incision at the 8th intercostal space was chosen. The pleural cavity appeared to be filled with dark, malodorous fluid. The fluid was sampled for pathological and microbiological examination and evacuated (nearly 1000 mL in total). Underneath the pleural effusion, the lung appeared widely covered with a thick layer of fibrin that tenaciously adhered to the underlying parenchyma; thus, the lung needed to be thoroughly decorticated. The postoperative course was uneventful apart from a prolonged air leak that lasted 6 days; the chest tube was removed on postoperative day 8 and the patient was discharged on postoperative day 10. During the hospital stay, the patient was administered levofloxacin 500 mg bid and Imipenem 1 g tid intravenously while waiting for the microbiological analysis results. Analgesia was carried out with acetaminophen 1 g tid and ketorolac 30 mg bid. Oral morphine was administered upon request at a maximum of 30 mg per day. At the histopathologic exam, an acute inflammation of the pleura with fibrin and reactive fibroblastic tissue was shown. Cultures showed the growth of *Actinomyces meyeri* colonies sensitive to imipenem and amoxicillin. The patient was therefore switched to imipenem 1 g tid intravenously alone for 4 days, discharged home from the hospital and advised to take oral amoxicillin 2 g tid for 12 months. The CRP dropped to 0.7 mg/dL. At discharge, his labs and ABG were within a normal range. The patient was followed in an outpatient setting for a week, two weeks, one month, and three months after discharge. Every time a chest X-ray was performed, it showed a gradual but constant improvement in the radiological appearance. At 12 months of follow-up, the chest X-ray showed a complete resolution of the left pleural effusion with complete re-expansion of the left lung (Figure 2).

## 3. Discussion

*Actinomycetes* are Gram-positive anaerobic bacteria belonging to the human commensal flora of the oropharynx, gastrointestinal tract, and urogenital tract. At the level of the oral cavity, they grow as obligatory or facultative anaerobes on periodontal lesions and carious teeth. These organisms may produce infection after local trauma, surgery, or aspiration. The main forms of actinomycosis are cervicofacial, thoracic, and abdominal [1,2,3,4]. *Actinomyces israelii* is the most frequently isolated in human infections, while other *Actinomyces* species are only occasionally implicated [4,5]. *Actinomyces meyeri* is a Gram-positive anaerobic bacterium that is typically found in the oral cavity and gastrointestinal tract. It can cause a range of infections, including abscesses, pneumonia, and endocarditis. However, it is most commonly associated with actinomycosis, a chronic infection that can cause the formation of abscesses and draining sinus tracts. The usual treatment of actinomycosis typically involves a prolonged course of antibiotics, usually penicillin or amoxicillin. In some cases, surgical drainage of abscesses may be necessary. The duration of treatment can range from several weeks to several months, depending on the severity of the infection and the response to treatment. It is important to complete the full course of antibiotics to ensure that the infection is fully eradicated. In some cases, antibiotic resistance may develop, and alternative antibiotics may be necessary. It is important to consult with a healthcare professional for proper diagnosis and treatment of actinomycetes and *Actinomyces meyeri* infections.

Traumatic pleural effusion is a common occurrence following chest trauma. It is usually a self-limiting condition that resolves without any significant intervention. The diagnosis is usually radiological with a chest X-ray, which can show the presence of fluid in the pleural space that appears as a white area. In addition, chest X-rays give information about the size and location of the effusion. In some cases, additional imaging tests, such as CT scans or ultrasounds, may be needed to confirm the diagnosis. In rare cases, pleural effusion can progress to pleural empyema, which is a serious and potentially life-threatening complication.

We have described a rare case of pleural empyema caused by *A. meyeri*, treated with antibiotic systemic therapy and surgical treatment with a thoracoscopic approach. The peculiarity of this report was the delayed diagnosis of Actinomyces pleuro-pulmonary infection in a patient who underwent recent chest trauma. *Actinomyces meyeri* differs from other actinomyces for its lack of filamentous rods and its predisposition to dissemination. The diagnosis of actinomycosis is made through the bacteriological identification of *Actinomyces*, but this occurs only in a minority of cases due to previous antibiotic therapy, inhibition of the growth of Actinomyces by concomitant and/or contaminating microorganisms and the conditions of inadequate culture or inadequate short-term incubation. For these reasons, the 16S rRNA sequencing techniques are fundamental for the detection of *Actinomyces* in clinical material. In this case report, ortho-panoramic radiography showed parodontopathy with important periodontal bone resorption, from which the *A. meyeri* infection probably spread to the lung and the pleura. Dentogingival disease is one of the major risk factors predisposing patients to lung infections due to the aspiration of the buccal flora, including *A. meyeri.* Thoracic or systemic trauma can be another predisposing factor for the infection, as well as chronic diseases, alcoholism, and the chronic use of immunosuppressants [2,3,4,5,6]. Here, the trauma due to the car accident, together with the parondontopathy, could represent a predisposing factor for the development of the infection, while alcohol abuse was not significant. Contextually, chest trauma turns out to be a misleading condition that contributed to a delayed correct diagnosis of *Actinomycetes* infection. Most cases of *A. meyeri* infections reported in the literature are associated with dentogingival disease and skin abscesses, but the spectrum of the infection seems to be wider, including brain and medullar lesions, breast and liver recurrent abscesses, and an almost constant involvement of the lung and pleura [2,3,4,5,6,7,8]. *Actinomyces meyeri* often causes disseminated disease, which can be secondary to the pulmonary infection [2,3]. In many cases, pneumonia is reported along with empyema and secondary spreading of the infection to the bones and the pericardium. The surgical treatment of the infection with drainage, sequestrectomy, or lobectomy is also reported in the literature as being associated with prolonged antibiotic therapy (in most cases with penicillin or amoxicillin for 6–12 months) [2,3,4,5,6,7,8,9,10,11]. Infections by *A. meyeri* are rare; however, they are probably underdiagnosed mainly because their identification has historically been difficult and unreliable [2,3,4]. Although rare, *Actinomycetes* infections must be considered, especially in cases of empyema or severe pneumonia with secondary systemic transmission, because in the majority of cases, the restitutio ad integrum is possible and complete with the correct pharmacological and surgical treatment [2]. *Actinomycetes* infections should also be considered in the differential diagnosis of pulmonary diseases such as lung cancer, lung abscesses, or pleural tuberculosis [2,3]. Actinomycosis is usually treated with penicillin G., although tetracyclines, erythromycin, clindamycin, and cephalosporins can also be used. While *A. meyeri* is not sensitive to fluoroquinolones, these are widely used for pulmonary infections. In cases of severe, organized pleural effusion with lung collapse, surgery is necessary [2,9,12]. In conclusion, *actinomycetes* infections must be considered in the face of non-solving empyema or severe pneumonia of unknown causes; in our case, this unexpected finding occurred in a patient with a supposed pleuropulmonary complication of chest trauma. In this case report, video-thoracoscopy was performed with complete pleural evacuation and lung decortication in addition to the pharmacological therapy, resulting in an almost complete resolution of the infection.

## Figures and Tables

**Figure 1 life-13-01450-f001:**
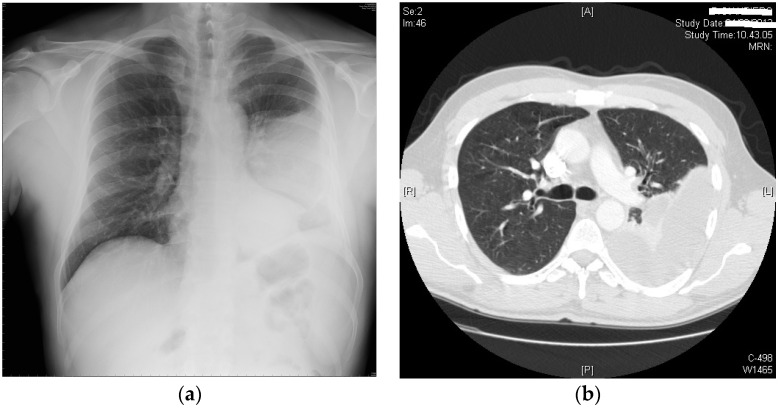
(**a**) Chest X-ray showing thickening of the left pleura and a pleural effusion; (**b**) CT scan at admission: organized pleural effusion with left lower lobe partial collapse.

**Figure 2 life-13-01450-f002:**
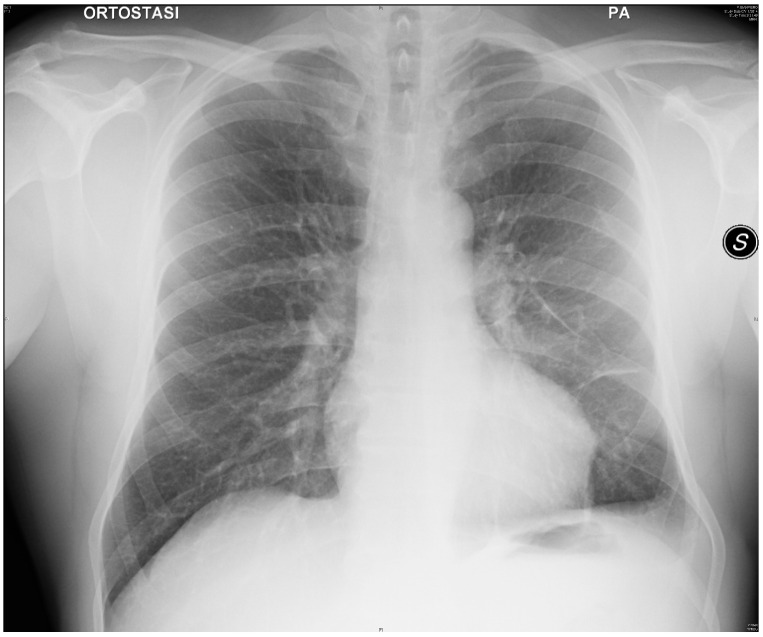
Twelve months follow-up chest X-ray showing resolution of the disease.

## Data Availability

The datasets used and analyzed during the current study are available from the corresponding author on reasonable request.

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
