# Peer review of "A Strange Case of Traumatic Pleural Effusion: Pleural Empyema Due to Actinomyces meyeri, a Case Report"

_life, 2023, doi:10.3390/life13071450_

Round 1
Reviewer 1 Report
The case was described and presented in detailed. However, some defects should be addressed.
(1) The manuscript are too lengthy and should be simplified and streamlined.
(2) Some terms were misused or meaning ambiguous, e.g. minor rib fracture , 800 cc. (please use ml.) of no longer clotting, case misunderstood etc. There are some more in the entire text. Please revise the whole manuscript by native professionals before the next submission.
(3) Why did you use a 18F catheter for blood drainage? It was obvious too small.
(4) At the first admission , the lung was only partially expanded after the thoracentesis. Why further intervention was not performed to make sure a full expansion ?
Some terms were misused. Already shown on the above box.
The entire manuscript are too lengthy, should be simplified and revised by native professionals.
Author Response
(1) The manuscript are too lengthy and should be simplified and streamlined.
I do agree. Unfortunately, the length was improved in response to a precise request from the editors at the very beginning of the submission. Therefore, some considerations tend to be redundant.
(2) Some terms were misused or meaning ambiguous, e.g. minor rib fracture , 800 cc. (please use ml.) of no longer clotting, case misunderstood etc. There are some more in the entire text. Please revise the whole manuscript by native professionals before the next submission.
Done. Everything seems fine now.
(3) Why did you use a 18F catheter for blood drainage? It was obvious too small.
I completely agree with you on that. The choice was driven by the scarce confidence of our pulmonologist with large-bore catheters.
(4) At the first admission , the lung was only partially expanded after the thoracentesis. Why further intervention was not performed to make sure a full expansion?
Again, this has to do with the fact that the first management was held in a secondary-level hospital without the prompt availability of the thoracic surgeon.
Reviewer 2 Report
Given that Actinomyces meyeri infections are rarely described in the literature, I consider this case report worthy of publication.
For a better overview of the case description, I suggest that the "Patient information" section be divided and marked with subheadings.
It is necessary to standardize the way of citations throughout the text.
The English language is well understood and written. Minor editing is required.
Author Response
Given that Actinomyces meyeri infections are rarely described in the literature, I consider this case report worthy of publication.
Thank you for your kind interest.
For a better overview of the case description, I suggest that the "Patient information" section be divided and marked with subheadings.
Thank you for the suggestion. This can be done only by the editors, if they agree.
It is necessary to standardize the way of citations throughout the text.
Again, thank you for the advice. As mentioned above, this is a matter to be addressed by the editors.
Reviewer 3 Report
Thank you for presenatation of your case report.
I do not have additional comment or question.
Author Response
Thank you for the nice review and for showing interest in our case-report.
Round 2
Reviewer 1 Report
One more suggestion is the word 'misunderstood' in the Abstract is inappropriate. I think 'underdiagnosis' is much better. Otherwise you have already revised the manuscript accordingly.
Author Response
One more suggestion is the word 'misunderstood' in the Abstract is inappropriate. I think 'underdiagnosis' is much better. Otherwise you have already revised the manuscript accordingly.
Thank you again for the suggestion. I'll change that immediately.